# Exploring the crosstalk between the FGF/FGFR pathway and tumor microenvironment in clear cell renal cell carcinoma

Takafumi Narisawa [1]*, Sei Naito[1], Yoshihide Mitsuda[2], Rintaro Ohe [3],
Hidenori Sato[4], Chizuru Kobayashi [5], Yuki Miyano[4], Hiromi Ito[1], Mitsuru Futakuchi [3],
Norihiko Tsuchiya[1]

1 Department of Urology, Faculty of Medicine, Yamagata University, Yamagata, Japan, 2 Medical Headquarters, Eisai Co., Ltd., Tokyo, Japan, 3 Department of Pathology, Faculty of Medicine, Yamagata University, Yamagata, Japan, 4 Division of Multi-omics Research, Yamagata University Well-Being Institute, Yamagata, Japan, 5 Deep Human Biology Learning, Eisai Co., Ltd., Tokyo, Japan

* tnari_0623@yahoo.co.jp

## Abstract

### Background

In the phase 3 CLEAR study, lenvatinib plus pembrolizumab showed improved efficacy versus sunitinib for patients with clear cell renal cell carcinoma (ccRCC). Previous preclinical studies demonstrated that lenvatinib attenuated tumor-associated macrophage (TAM) infiltration into tumor tissues by inhibiting fibroblast growth factor receptor (FGFR). However, the role of the FGFR pathway in ccRCC remains underexplored. This study aims to evaluate FGFR1–4 expression in ccRCC and investigate its relationship with the tumor microenvironment, particularly TAM.

### Methods

We primarily analyzed FGFR1–4 expression and CD163 positive cell count as estimation of TAM infiltration in 57 ccRCC specimens from patients undergoing nephrectomy using immunohistochemistry. Transcriptomic analysis was performed to assess immune-related gene signature and gene expressions.

### Results

FGFR1 expression was elevated in over 80% of ccRCC samples and was significantly associated with increased CD163-positive TAM infiltration. FGFR1 expression was also negatively correlated with the IMmotion150 Teff gene signature and the expression of interferon-γ signaling targeted genes such as *IFNG*, *GZMB*, and *CD274*, suggesting an immunosuppressive phenotype. In contrast, FGFR2 and FGFR4 expression were less prevalent, and FGFR3 expression was not detected.

**Data availability statement:** The data underlying this study cannot be made publicly available due to ethical and legal restrictions related to patient privacy. Data are available from the Ethics Committee of Yamagata University Faculty of Medicine (contact via email: yu-ikekenkyu@jm.kj.yamagata-u.ac.jp) for researchers who meet the criteria for access to confidential data.

**Funding:** Eisai Co., Ltd. provided research funding under a collaborative research agreement with Yamagata University. The funder was involved in the study design, data collection and analysis, and preparation of the manuscript, but the final decision regarding the content and submission of the manuscript was made by the academic authors. TN, SN, and NT received honoraria from Eisai Co., Ltd. YM and CK are employees of Eisai Co., Ltd. RO, HS, YM, HI, and MF have no financial conflicts of interest to disclose. This does not alter our adherence to PLOS ONE policies on sharing data and materials.

**Competing interests:** The authors have declared that no competing interests exist.

## Conclusions

This study provides the first comprehensive evaluation of FGFR1–4 expression in ccRCC and suggests that FGFR1 expression may contribute to the immunosuppressive tumor microenvironment by recruiting TAM. These findings indicate that FGFR1 could serve as a potential biomarker for therapeutic strategies and highlight the need for further research to explore FGFR-targeted therapies in ccRCC.

## Introduction

Renal cell carcinoma (RCC) ranks as the 14th most prevalent malignancy worldwide, with over 434,000 new cases reported in 2022 [1]. Systemic treatments for metastatic RCC (mRCC) have advanced significantly, including the development of tyrosine kinase inhibitors (TKIs), mammalian target of rapamycin (mTOR) inhibitors, immune checkpoint inhibitors (ICIs), and ICI-based combinations, all of which have contributed to improved overall survival [2,3]. Clear cell RCC (ccRCC), accounting for approximately 75% of RCC cases, has traditionally been treated with vascular endothelial growth factor receptor (VEGFR)-TKIs [4]. Inactivation of the tumor suppressor gene *VHL*, which upregulates genes that promote cellular proliferation and angiogenesis such as vascular endothelial growth factor (VEGF), is a key mechanism in the development of ccRCC [5]. Consequently, the biological rationale for VEGFR inhibition in ccRCC is well understood. However, the role of the fibroblast growth factor receptor (FGFR) in the development and maintenance of ccRCC has not been thoroughly investigated. Previous studies have demonstrated that FGFR1 is expressed in 32.3–98% of primary renal tumors, while FGFR2 expression is observed in 4–66.2% of cases [6,7]. Increased FGFR1 expression has been correlated with decreased progression-free survival in patients with mRCC receiving sorafenib [8]. Additionally, activation of the FGFR pathway may contribute to resistance against VEGFR-targeted therapies [9]. Whole-genome analyses of ccRCC revealed gene copy number gains of FGFR4 in 65% of patients [10]. Furthermore, 54% of ccRCC cases exhibited elevated FGFR4 expression, and treatment with BLU9931, a selective FGFR4 inhibitor, significantly reduced tumor growth in ccRCC xenograft models [11]. In the phase 3 CLEAR study, the combination of lenvatinib, multi-TKI targeting VEGFR1–3, FGFR1–4, C-KIT, RET, and PDGFR-α, along with pembrolizumab, demonstrated significant clinical benefits for patients with mRCC [12]. These outcomes suggest that FGFRs could serve as potential therapeutic targets in ccRCC. As combination immunotherapies have become the standard of care in first-line treatment for mRCC, understanding the alternative roles of drug-targeted factors in the tumor microenvironment is crucial. The infiltration of anti-inflammatory macrophages in ccRCC has been associated with increased malignancy and poorer prognosis [13,14]. Biomarker analyses from the IMmotion150 study identified myeloid inflammation as one of the mechanisms of resistance to ICI-combination therapy in ccRCC [15]. Preclinical studies showed that FGFR inhibition by lenvatinib reduced tumor-associated macrophages (TAMs) and activated CD8-positive T cells in an interferon-γ (IFNγ)-dependent manner [16,17].

Moreover, it has been reported that TAMs express FGFR1 and FGFR2 [18]. Although immunomodulatory effects of FGFR signaling have been suggested, the role of the FGFR pathway in the tumor microenvironment of ccRCC remains unknown. In this study, we evaluated the immunohistochemical (IHC) expression of FGFR1–4 in ccRCC patients and investigated the relationship between FGFRs and the intra-tumor microenvironment, with a focus on macrophage infiltration.

## Materials and methods

### Clinical specimens and ethics

We selected 57 patients with mRCC who had undergone cytoreductive nephrectomy at Yamagata University between 2009 and 2020, and for whom complete package of clinical information and well-preserved specimens were available. This study was approved by the Ethical Committee of Yamagata University (approval number: 2021−2) based on the tenets of the Declaration of Helsinki, and written informed consent for the use of clinical specimens was obtained from all patients.

### Immunohistochemistry

We examined specimens of patients who were histologically diagnosed to have ccRCC by a pathologist. IHC staining and slide preparation were performed as previously described [11]. Single immunostainings were performed with BOND RXm (Leica Biosystems, Nussloch, Germany) according to the manufacturer's protocol.

We used antibodies of CD163 (10D6, Leica Biosystems), FGFR1 (M17D10, Novus), FGFR2 (D4L2V, CST), FGFR3 (4574S, CST), FGFR4 (HPA027369, Sigma-Aldrich) to evaluate the number of macrophage and that of stromal tissue in ccRCC specimens.

For quantitative analysis, whole-slide imaging of the IHC slides was generated with a virtual scanner (NanoZoomer, HAMAMATSU, Hamamatsu, Japan). To demonstrate the distribution of macrophage, HALO software (Indica Labs, Albuquerque, NM, USA) was used to allocate positive cells on a grid chart of section. The number of positive cells and that of total nucleated cells was counted in the annotated tumor area.

Two experienced physicians visually evaluated the staining intensity of FGFRs using a four-point scale (negative—no staining: 0 point, weak—stains to the same degree as: 1 point, moderate—stains stronger: 2 point, strong—stains considerably stronger: 3 point) according to our previous study [11], and IHC scores were calculated as the mean point of three different areas per slide to account for intratumoral heterogeneity.

### Next generation sequence and transcriptome analysis

A library was constructed using the extracted RNA with Ion AmpliSeq Transcriptome Human Gene Expression Kit (A26325, Thermo Fisher scientific) according to the manufacturer's protocol. Amplicon-based library for each transcript was amplified from extracted RNAs specimens by RT-PCR. Quantification of libraries was carried out a 4150 TapeStation system (Agilent Technologies) using a D1000 ScreenTape (5067–5582, Agilent Technologies). Prepared libraries were performed on semi-conductor sequencer by Ion GeneStudio S5 (Thermo Fisher scientific) using Ion 540 Kit-OT2 and Ion 540 Chip (A27753, A27765, Thermo Fisher scientific). Sequenced data was analyzed using in house STAR-RSEM nextflow pipeline using high performance computing system. The obtained sequenced reads were mapped with the human genome reference build GRCh38/hg38 by STAR program (https://github.com/alexdobin/STAR,Ver.2.7.10a), and quantification of gene transcripts per million (TPM) were calculated by RSEM program (https://github.com/deweylab/RSEM, Ver.1.3.3). TPM filtered out low-expressed genes, and third quantile normalization was used for subsequent analysis.

### Gene signature analysis

IMmotion150 Teff gene signature was calculated as previously described [15]. To calculate score for the signature, the expression level for each gene was standardized by first determining the mean expression level (log TPM) and standard

deviation of that gene within the entire evaluable population across both study arms and then subtracting the mean value and dividing by the standard deviation for expression. The pathway score was then calculated as the mean of the standardized genes in the pathway.

### Endpoints

The primary endpoints of this study were to observe FGFRs expression in ccRCC specimens and to identify the correlations between FGFRs expression and intra-tumoral CD163-positive cells as a marker for TAMs. Secondary endpoints were to show the associations between FGFRs expression, IMmotion150 Teff signature, and IFNγ-mediated genes. Exploratory, we investigated the correlation of FGFRs expression and survivals, patient backgrounds, and clinical outcomes of nivolumab treatment.

### Statistical analysis

All statistical analyses were performed using R (version 4.1.0). A Welch's t test was used to analyze the differences between two groups. Survival curves were plotted using the Kaplan–Meier method and a log-rank test was used to compare two groups. The relationship between FGFRs expression and clinicopathological features was analyzed by the Fisher's exact test. The correlation between IHC scores of FGFRs and IFNγ-mediated genes was estimated using Pearson's method. A two-sided $p$-value <0.05 was considered statistically significant.

## Results

### Characteristics of metastatic ccRCC patients

We selected 57 cases of metastatic ccRCC treated at Yamagata University Hospital, from which comprehensive analyses of whole transcriptome data and IHC staining were possible using primary tumor specimens. The median follow-up period was 47.7 months (range: 3.2–148.5 months). The mean age at RCC diagnosis was 65.4 years (range: 42.6–85.0 years), and 87.7% of the patients were male. Detailed clinical information, including IMDC risk classification, clinical T stage, WHO grade, presence of primary metastasis, and metastatic sites, was collected and summarized in Table 1.

### Protein expression of FGFRs in ccRCC clinical specimens

We assessed protein expression levels of FGFR1, FGFR2, FGFR3, and FGFR4 in 57 surgical specimens of ccRCC using IHC staining. Among these, FGFR1, FGFR2, and FGFR4 were detected in tumor tissues, whereas FGFR3 expression was absent in all cases (Fig 1A–D). The median IHC scores for FGFR1, FGFR2, and FGFR4 were 2.33, 0.33, and 1.33, respectively (Fig 1E). When applying a cut-off IHC score of 2.00, high expression levels were observed in 46 cases (80.7%) for FGFR1, 13 cases (22.8%) for FGFR2, and 8 cases (14.0%) for FGFR4 (Fig 1F).

### Association between FGFR expression and clinicopathological characteristics

The association between FGFR expression levels and various clinicopathological characteristics is summarized in Table 2. There was a trend that high expression of FGFR4 was associated with both synchronous metastasis (Odds ratio [95%CI] = 7.29 [0.83–63.79], $p = 0.059$) and lymph node metastasis (Odds ratio [95%CI] = 5.13 [1.06–24.87], $p = 0.052$). In contrast, FGFR1 and FGFR2 expression showed no association with these metastatic factors.

Among the IMDC risk factors, platelet count ($p = 0.035$) and serum corrected calcium levels ($p = 0.048$) were significantly associated with high FGFR1 expression, whereas serum corrected calcium levels ($p = 0.012$) and neutrophil count ($p = 0.028$) were significantly associated with low FGFR2 expression (S1 Table). Furthermore, patients whose tumors exhibited high FGFR2 expression demonstrated significantly longer cancer-specific survival ($p = 0.045$). FGFR1 ($p = 0.880$) and FGFR4 ($p = 1.000$) expression levels, however, were not significantly associated with survival outcomes (S1A–C Fig).

**Table 1. Patient characteristics for the analyzed ccRCC specimens.**

| Characteristic | Total, n = 57 |
|---|---|
| Observation time, months [range] | 47.7 [3.2–148.5] |
| Age, median [range] | |
| at diagnosis of RCC | 65.4 [42.6–85.0] |
| at diagnosis of metastasis | 66.5 [44.1–85.7] |
| Sex, n (%) | |
| Male | 50 (87.7) |
| Female | 7 (12.3) |
| IMDC risk classification, n (%) | |
| Favorable | 10 (17.5) |
| Intermediate | 39 (68.4) |
| Poor | 5 (8.8) |
| Not evaluated | 3 (5.3) |
| Clinical T stage, n (%) | |
| T1 | 22 (38.6) |
| T2 | 8 (14.0) |
| T3 | 23 (40.4) |
| T4 | 4 (7.0) |
| Fuhrman grade, n (%) | |
| 1 | 3 (5.3) |
| 2 | 19 (33.3) |
| 3 | 29 (50.9) |
| 4 | 6 (10.5) |
| Primary metastasis, n (%) | |
| Yes | 31 (54.4) |
| No (metachronous) | 26 (45.6) |
| Site of metastasis, n (%) | |
| Lung | 35 (61.4) |
| Bone | 13 (22.8) |
| Lymph node | 12 (21.1) |
| Liver | 2 (3.5) |

## Macrophage infiltration in ccRCC with elevated FGFR1 expression

To elucidate the impact of FGFR expression on macrophage infiltration within ccRCC tumors, we analyzed TAMs by quantifying CD163-positive cells as a marker for TAMs [13]. Our findings indicate that TAM infiltration was significantly increased in tumors with elevated FGFR1 expression ($p = 0.048$) (Fig 2A–B). Conversely, no significant relationship was observed between TAM infiltration and the expression of FGFR2 ($p = 0.159$) or FGFR4 ($p = 0.967$).

## Immunosuppressive phenotype in FGFR1-high tumors

To investigate the immunosuppressive activity of TAMs in ccRCC with high FGFR1 expression, we assessed various immunological markers and gene signatures. Our analysis demonstrated that the IMmotion150 Teff signature was significantly lower in tumors with high FGFR1 expression ($p = 0.026$) (Fig 3A). Furthermore, a negative correlation was observed between FGFR1 IHC scores and the expression levels of IFNγ-targeted genes, including *IFNG* (correlation coefficient ($r$) = −0.30, $p = 0.025$), *GZMB* ($r = -0.29$, $p = 0.027$), and *CD274* ($r = -0.30$, $p = 0.025$) (Fig 3B), suggesting that higher FGFR1

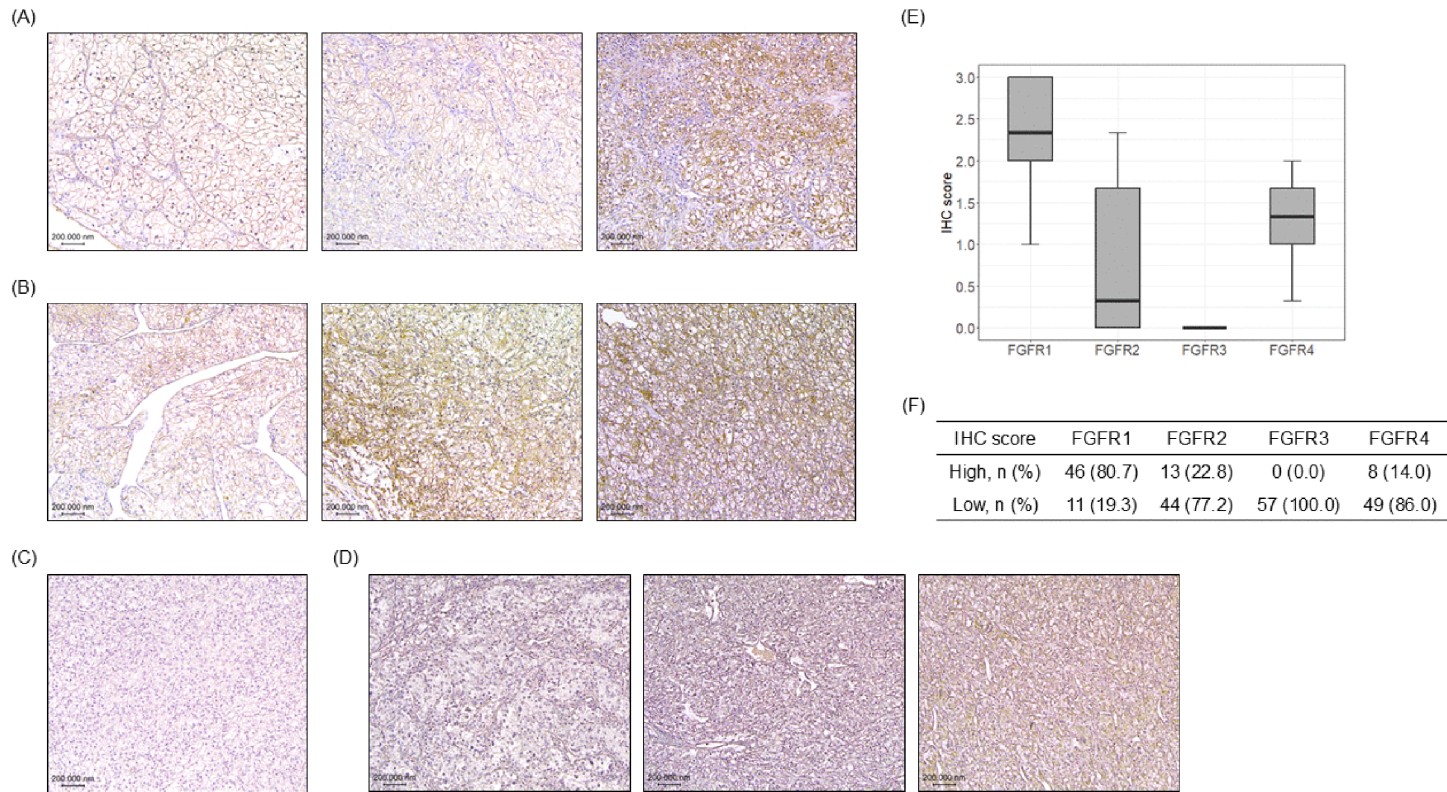

**Fig 1. FGFR1–4 protein expression analysis in ccRCC clinical specimens.** IHC evaluation of FGFR1–4 subtype expression in clinical samples of ccRCC. IHC images are presented sequentially from low to high expression levels, arranged from left to right. (A) Representative image of FGFR1 staining. (B) Representative image of FGFR2 staining. (C) Representative image of FGFR3 staining. Notably, no specimens demonstrated FGFR3 expression. (D) Representative image of FGFR4 staining. (E) The distribution of IHC scores for each FGFR subtype across the analyzed ccRCC samples. (F) Expression profile of FGFRs, with a cut-off value set at an IHC score of 2.00, demonstrating the proportion of samples with expression levels above and below this threshold.

expression may be linked to reduced IFNγ-related immune activity. In contrast, FGFR2 and FGFR4 IHC scores did not show significant correlations with the IMmotion150 Teff signature or these IFNγ-targeted genes (S2A–B Fig). Additionally, we noted a trend indicating reduced efficacy of nivolumab monotherapy across all treatment lines in patients exhibiting high FGFR1 expression (Fig 3C and S2C Fig). This might suggest a potential resistance to ICIs in FGFR1-high tumors.

## Discussion

To our knowledge, this is the first study to evaluate FGFR expression in relation to the immune microenvironment in ccRCC. We assessed the expression levels of FGFR1–4 in ccRCC patients and investigated their association with the intra-tumoral microenvironment.

Notably, more than 80% of ccRCC patients in our cohort exhibited high FGFR1 expression, a finding that aligns with previous studies [6]. One plausible explanation for the widespread expression of FGFR1 in ccRCC is the frequent occurrence of *VHL* mutations in these patients. Loss of functional VHL protein results in abnormal accumulation of FGFR1 on the cell surface, which may contribute to the high FGFR1 expression observed in ccRCC tumors [19]. This accumulation could potentially alter the tumor microenvironment by promoting immunosuppressive conditions that support tumor progression.

**Table 2. Association between FGFR subtype expressions with clinical and pathological factors.**

| Variable | | FGFR1 | | | | FGFR2 | | | | FGFR4 | | | |
|---|---|---|---|---|---|---|---|---|---|---|---|---|---|
| | n | High | Low | Odds ratio [95%CI] | p | High | Low | Odds ratio [95%CI] | p | High | Low | Odds ratio [95%CI] | p |
| IMDC risk classification | | | | | | | | | | | | | |
| Favorable | 10 | 8 | 2 | | 0.745 | 4 | 6 | | 0.275 | 1 | 9 | | 1.000 |
| Intermediate | 39 | 30 | 9 | 0.83 [0.15–4.65] | | 9 | 30 | 0.45 [0.10–1.95] | | 6 | 33 | 1.64 [0.17–15.40] | |
| Poor | 5 | 5 | 0 | 3.24 [0.13–80.99] [†] | | 0 | 5 | 0.13 [0.01–3.02] [†] | | 1 | 4 | 2.25 [0.11–45.72] | |
| Clinical T stage | | | | | | | | | | | | | |
| T1 or T2 | 30 | 25 | 5 | | 0.740 | 9 | 21 | | 0.216 | 4 | 26 | | 1.000 |
| T3 or T4 | 27 | 21 | 6 | 0.70 [0.19–2.62] | | 4 | 23 | 0.41 [0.11–1.52] | | 4 | 23 | 1.13 [0.25–5.04] | |
| Fuhrman grade | | | | | | | | | | | | | |
| 1 or 2 | 22 | 17 | 5 | | 0.733 | 5 | 17 | | 1.000 | 2 | 20 | | 0.466 |
| 3 or 4 | 35 | 29 | 6 | 1.42 [0.38–5.37] | | 8 | 27 | 1.01 [0.28–3.59] | | 6 | 29 | 2.07 [0.38–11.31] | |
| Primary metastasis | | | | | | | | | | | | | |
| Yes | 31 | 25 | 6 | 0.99 [0.26–3.72] | 1.000 | 7 | 24 | 0.97 [0.28–3.36] | 1.000 | 7 | 24 | 7.29 [0.83–63.79] | 0.059 |
| No (metachronous) | 26 | 21 | 5 | | | 6 | 20 | | | 1 | 25 | | |
| Lymph node metastasis | | | | | | | | | | | | | |
| Yes | 12 | 9 | 3 | 0.65 [0.14–2.95] | 0.683 | 2 | 10 | 0.62 [0.12–3.26] | 0.713 | 4 | 8 | 5.13 [1.06–24.87] | 0.052 |
| No | 45 | 37 | 8 | | | 11 | 34 | | | 4 | 41 | | |

[†]; Modified odds ratio: For frequencies equal to 0, 0.5 was added to each to calculate odds ratios.

(A)

(B)

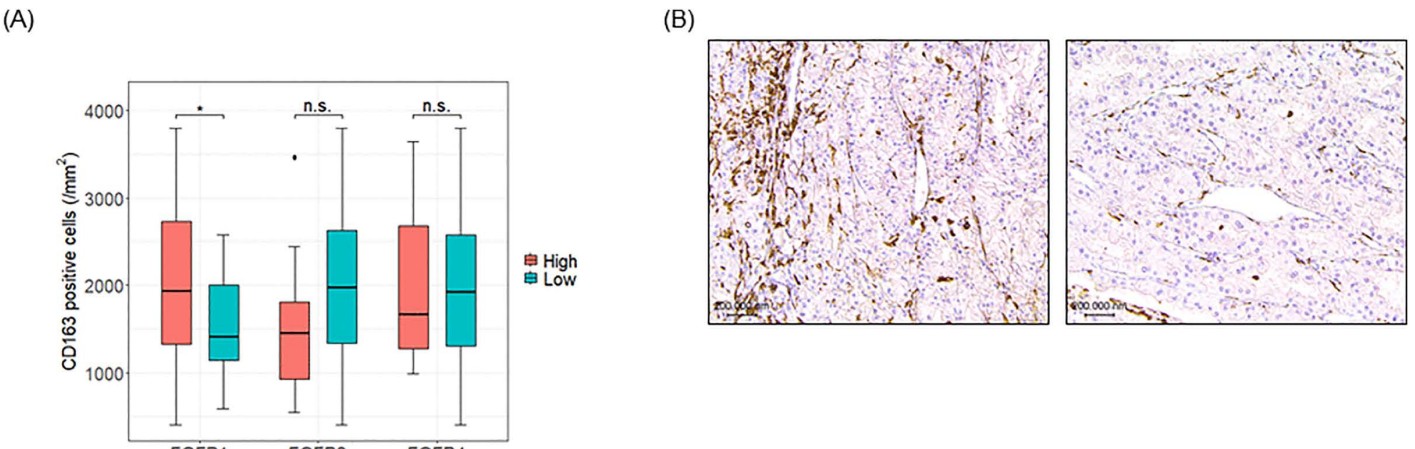

**Fig 2. Analysis of FGFR subtype expression levels and intra-tumor macrophage infiltration.** IHC evaluation of intra-tumor macrophage infiltration across FGFR subtypes. (A) Differential expression of CD163-positive cells across FGFR subtypes. The number of CD163-positive cells per unit area was significantly higher in the high FGFR1 expression group compared to the low FGFR1 expression group, indicating increased macrophage infiltration in tumors with elevated FGFR1 expression. *; $p<0.05$, n.s.; not significant. (B) Representative images of CD163-positive cells in tumors with differing FGFR1 expression levels. The left image shows a specimen with high FGFR1 expression, while the right image depicts a specimen with low FGFR1 expression, illustrating the difference in macrophage infiltration.

In this study, approximately 20% of ccRCC patients exhibited high expression of FGFR2. The reported expression levels of FGFR2 in ccRCC, however, vary across studies [6,7] These findings highlight a distinct feature in the Japanese ccRCC cohort, which is characterized by relatively high expression of FGFR1 and, to a lesser extent, FGFR2.

(A)

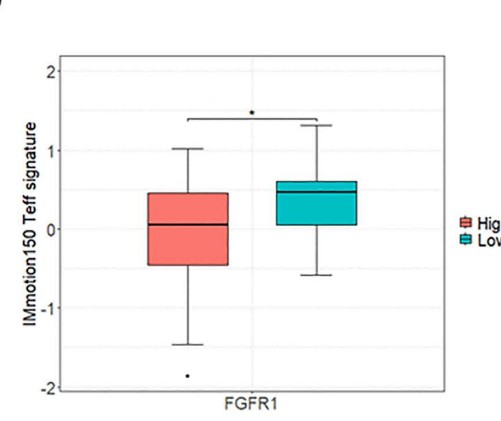

(B)

| Variable | r [95%CI] | p |
|---|---|---|
| IFNG | -0.30 [-0.52 to -0.04] | 0.025 |
| GZMB | -0.29 [-0.51 to -0.04] | 0.027 |
| CD274 | -0.30 [-0.52 to -0.04] | 0.025 |

(C)

| | High (n = 17) | Low (n = 4) |
|---|---|---|
| Objective response rate, % | 29.4 | 50.0 |
| Best response, n (%) | | |
| CR | 1 (5.9) | 0 (0.0) |
| PR | 4 (23.5) | 2 (50.0) |
| SD | 4 (23.5) | 0 (0.0) |
| PD | 8 (47.1) | 2 (50.0) |

**Fig 3. Comparison of immunosuppressive phenotypes and ICI treatment response in relation to FGFR1 expression.** Transcriptome analysis of immune-related genes and evaluation of clinical response of nivolumab. (A) Decreased expression of the IMmotion150 Teff signature in tumors with high FGFR1 expression, indicating a potential immunosuppressive phenotype associated with elevated FGFR1 levels. *; $p < 0.05$. (B) Correlation between FGFR1 IHC score and IFNγ signaling targeted genes using Pearson's correlation analysis. A negative correlation was observed between FGFR1 expression and the transcript levels of IFNγ-targeted genes including *IFNG*, *GZMB*, and *CD274*. (C) Objective response to nivolumab across all treatment lines, comparing differences in FGFR1 expression levels. CR; complete response, PR; partial response, SD; stable disease, PD; progressive disease.

The role of FGFR3 in the pathogenesis of RCC has been previously investigated, with one study assessing *FGFR3* mutational activity in 238 primary renal tumor samples, where no alterations in *FGFR3* were identified [20]. In our study, IHC expression of FGFR3 was similarly undetected, supporting the notion that FGFR3 is unlikely to play a role in RCC development.

Copy number amplification and elevated protein expression of FGFR4 have been reported in 59.5% of ccRCC cases, suggesting a potential role for FGFR4 in ccRCC pathogenesis [11]. In our study, high expression of FGFR4 was observed in 14.0% of patients. This finding, while lower than previously reported, indicates that FGFR4 may have pathological significance in ccRCC, particularly when considered in relation to clinicopathological factors.

Existing literature reveals that the prognostic value of FGFR expression varies across different cancer types and patient cohorts, with contradictory findings reported [21]. In RCC, for instance, FGFR2 protein overexpression has been identified as a negative prognostic marker for cancer-specific survival [7]. Another study in mRCC demonstrated that FGFR2 expression serves as an adverse prognostic factor in patients treated with targeted agents [22]. However, in our study, despite the relatively small patient cohort, higher protein expression of FGFR2 was significantly associated with better prognosis in ccRCC, while FGFR1 and FGFR4 expression showed no association with cancer-specific survival. Possible reasons for the inconsistency may be that all patients in this study had metastatic disease and treatment modality differed among patients.

In this study, there was a trend that high FGFR4 expression was associated with clinicopathological features such as primary metastasis and lymph node metastasis. Similarly, in gastric cancer, elevated FGFR4 protein expression has been linked to distant metastasis and lymph node involvement [23]. Elevated mRNA expression of FGFR4 has been reported at recurrent or metastatic sites compared to primary tumors in adrenocortical carcinoma, underscoring a potential role for FGFR4 in promoting metastatic behavior [24]. Furthermore, activation of the FGFR4 pathway has been associated with distant metastasis across various cancer types, reinforcing its involvement in tumor progression [25–27].

Conversely, clinicopathological factors such as clinical T stage and Fuhrman grade, previously identified as prognostic indicators in various cancers by several studies [28–30], did not show significant differences in our analysis. Likewise, IMDC risk classification was not significantly associated with FGFR expression in our cohort, although individual IMDC risk factors including platelet count, corrected calcium, and neutrophil count demonstrated significant differences: these markers were elevated in patients with higher FGFR1 expression and reduced in those with lower FGFR2 expression. Collectively, these findings suggest that FGFR1 and FGFR4 may contribute to the malignant potential of RCC, particularly in the context of metastatic progression, while FGFR2 might confer a protective effect to some extent.

Previous studies across various cancers have suggested a possible correlation between FGFR pathway activation and macrophage infiltration. In gastric cancer, for instance, the extent of macrophage infiltration has been shown to correlate positively with FGFR1 expression in tumor cells [31]. Similarly, FGFR expression has been linked to M2 macrophage infiltration in triple-negative breast cancer [32], while in esophageal cancer, FGFR1 signaling was found to regulate the survival and migration of TAMs [33]. Additionally, activated FGFR1 signaling in mammary tumor cells has been shown to promote macrophage recruitment [34]. These findings collectively indicate a potential intra-tumor correlation between FGFR1 expression and TAM infiltration in also ccRCC. In our study, we observed a trend suggesting reduced efficacy of nivolumab in patients with higher FGFR1 expression and increased macrophage infiltration. This trend raises the possibility that FGFR1-associated macrophage recruitment may contribute to an immunosuppressive microenvironment, potentially diminishing the effectiveness of ICIs in ccRCC.

Molecular analyses from the IMmotion150 study align with this trend, indicating that the efficacy of atezolizumab, whether in combination with bevacizumab or as monotherapy, was reduced in patients with higher levels of myeloid inflammation [15]. Preclinical studies have similarly suggested that FGFR inhibition, such as with lenvatinib, may enhance antitumor immunity by reducing TAM infiltration [16,17]. These findings imply that FGFR signaling activation and TAM-mediated immunosuppression may contribute to resistance against ICIs. Thus, a therapeutic approach targeting macrophage recruitment via FGFR inhibition could be a potential strategy to address FGFR-mediated resistance to ICIs on ccRCC treatment, although further investigation is needed to clarify this approach's clinical relevance.

This study has certain limitations. It is a retrospective analysis based on older and small sample sizes. Additionally, we analyzed only primary tumor specimens and did not assess the molecular profiles of metastatic sites. Given the heterogeneity of ccRCC [35] and reports of discordance in FGFR1 protein levels between primary and metastatic sites [36], it is recommended to examine FGFR expression levels and tumor microenvironment in metastatic sites in future studies.

Previous research in hepatocellular carcinoma has suggested that FGFR4 protein levels may serve as a biomarker for lenvatinib monotherapy [37], indicating that FGFR expression in ccRCC could also potentially play a role in predicting responses to FGFR-targeted treatments. However, in this study, the limited number of patients who received systemic therapy constrained our ability to evaluate the correlation between FGFR expression levels and the efficacy of such therapies. To fully understand the clinical relevance of FGFR expression in ccRCC, further studies are needed. Such research could help clarify whether FGFRs might serve as useful biomarkers in guiding therapeutic decisions.

## Conclusion

In summary, our study represents the first comprehensive report to evaluate FGFR subtype expression in ccRCC through IHC analysis, examining its associations with the tumor microenvironment and clinicopathological factors. This analysis provides a foundational understanding of the potential roles of FGFRs in ccRCC pathogenesis and their relationship to clinical features.

## Supporting information

**S1 Table. Association between FGFR subtype expressions with IMDC risk factors.**
(TIF)

**S1 Fig. Comparison of FGFR Subtype differences in relation to overall survival.** Kaplan-Meier curves showing probability of overall survival across the expression levels of FGFR1 (A), FGFR2 (B), and FGFR4 (C). Sample numbers per group indicated below the graphs.
(TIF)

**S2 Fig. Comparison of immunosuppressive phenotypes and ICI treatment response in relation to FGFR2 and FGFR4 expression.** Transcriptome analysis of immune-related genes and evaluation of clinical response of nivolumab. (A) IMmotion150 Teff signature scores across the expression levels of FGFR2 and FGFR4. n.s.; not significant. (B) Correlation between IHC score of FGFR2 or FGFR4 and IFNγ signaling targeted genes using Pearson's correlation analysis. (C) Objective response to nivolumab across all treatment lines, comparing differences in FGFR2 and FGFR4 expression levels. CR; complete response, PR; partial response, SD; stable disease, PD; progressive disease.
(TIF)

## Author contributions

**Conceptualization:** Takafumi Narisawa, Sei Naito, Yoshihide Mitsuda, Norihiko Tsuchiya.

**Data curation:** Sei Naito, Chizuru Kobayashi.

**Formal analysis:** Sei Naito, Chizuru Kobayashi.

**Funding acquisition:** Yoshihide Mitsuda.

**Investigation:** Takafumi Narisawa, Rintaro Ohe, Hidenori Sato, Yuki Miyano, Hiromi Ito.

**Methodology:** Takafumi Narisawa, Sei Naito, Yoshihide Mitsuda.

**Project administration:** Takafumi Narisawa, Sei Naito, Yoshihide Mitsuda.

**Resources:** Takafumi Narisawa, Sei Naito, Yoshihide Mitsuda, Rintaro Ohe, Hidenori Sato, Yuki Miyano, Hiromi Ito.

**Software:** Chizuru Kobayashi.

**Supervision:** Hidenori Sato, Mitsuru Futakuchi, Norihiko Tsuchiya.

**Validation:** Chizuru Kobayashi.

**Visualization:** Chizuru Kobayashi.

**Writing – original draft:** Takafumi Narisawa, Yoshihide Mitsuda.

**Writing – review & editing:** Yoshihide Mitsuda.

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
