## [Decision Letter · Decision Letter 0]

31 Oct 2025

Dear Dr. Narisawa,

plosone@plos.org

We look forward to receiving your revised manuscript.

Kind regards,

Pirkko L. Härkönen, M.D., Ph.D.

Academic Editor

PLOS ONE

Journal Requirements:

3. In the online submission form, you indicated that “Access to the data may be granted upon reasonable request to the corresponding author, subject to approval by the institutional ethics committee of Yamagata University.”

Reviewers' comments:

Reviewer's Responses to Questions

**Comments to the Author**

1. Is the manuscript technically sound, and do the data support the conclusions?

Reviewer #1: Yes

Reviewer #2: Partly

2. Has the statistical analysis been performed appropriately and rigorously?

Reviewer #1: Yes

Reviewer #2: Yes

3. Have the authors made all data underlying the findings in their manuscript fully available?

Reviewer #1: Yes

Reviewer #2: Yes

4. Is the manuscript presented in an intelligible fashion and written in standard English?

Reviewer #1: Yes

Reviewer #2: Yes

Reviewer #1: The fibroblast growth factor receptor (FGFR) family has garnered increasing attention in oncology due to its multifaceted role in GU tumors biology. This manuscript provides a timely and comprehensive investigation into the interplay between FGFR signaling and the tumor microenvironment in clear cell renal cell carcinoma (ccRCC). The study is well-conceived and methodologically sound, offering new insights into the biological significance of FGFR1 within the renal cancer microenvironment - an aspect that has not been fully elucidated in previous work. The data are clearly presented, the analyses are rigorous, and the conclusions are well supported by the findings. Overall, this is an elegant and well-executed study that meaningfully advances current understanding of FGFR biology in ccRCC. I recommend the manuscript for publication in its present form.

Reviewer #2: Authors analyzed FGFR1-4 expression and CD163 positive cell count for estimation of TAM infiltration in 57 ccRCC specimens from patients undergoing nephrectomy using immunohistochemistry. This study indings indicate that FGFR1 could serve as a potential biomarker for therapeutic strategies and highlight the need for further research to explore FGFR-targeted therapies in ccRCC.

Authors described methods for quantification of IHC staining. “Staining of FGFRs was evaluated using a four-point scale (negative no staining: 0 point, weak stains to the same degree as: 1 point, moderate stains stronger: 2 point, strong stains considerably stronger: 3 point), and IHC scores were calculated as the mean point of three different areas per slide to account for intratumoral heterogeneity”

Q: However, it was not clear to me how they decide staining levels as “weak stains to the same degree as: 1 point, moderate stains stronger: 2 point, strong stains considerably stronger” for scoring? Please describe the method in more detail.

Authors commented no availability of tumor samples from metastatic sites. However, it is critical to examine FGFR expression levels even using separate sets of clinical samples in this study.

C: Please show results of FGFR IHC in metastatic sites in addition to the primary tumors. FGFR4 showed trends of association with primary metastasis, and I wonder if FGFR4 expressions were selectively high in metastatic sites. Especially author showed association of high FGFR4 expression in RCC patients with metastases

When applying a cut-off IHC score of 2.00, high expression levels were observed in 46 cases (80.7%) for FGFR1, 13 cases (22.8%) for FGFR2, and 8 cases (14.0%) for FGFR4 (Fig 1F)

Q: What is the rationale to use a cut-off IHC score of 2.00 for high vs low? There was no association of FGFR4 expression with survival, although there were trends of FGFR4 expression with synchronous metastasis and lymph node metastasis.

Analysis of FGFR subtype expression levels and intra-tumor macrophage infiltration

Q: Please confirm if intra-tumor macrophage expressed FGFR by co-staining with CD163.

**Do you want your identity to be public for this peer review?** For information about this choice, including consent withdrawal, please see our Privacy Policy

Reviewer #1: **Yes:** Ilya Tsimafeyeu

Reviewer #2: No

---

## [Author Response · Author response to Decision Letter 1]

11 Dec 2025

Please refer to the attached file named "Response to Reviewers.docx" for our detailed point-by-point responses.

We have uploaded the response as a separate document because it includes representative images (Reviewer Figure 1) addressing Reviewer #2’s comments regarding spatial relationships, which cannot be displayed within this text box.

---

## [Editor Report · Decision Letter 1]

15 Dec 2025

Exploring the crosstalk between the FGF/FGFR pathway and tumor microenvironment in clear cell renal cell carcinoma

PONE-D-25-37743R1

Dear Dr. Takafumi Narisawa,

We’re pleased to inform you that your manuscript has been judged scientifically suitable for publication and will be formally accepted for publication once it meets all outstanding technical requirements.

Kind regards,

Pirkko L. Härkönen, M.D., Ph.D.

Academic Editor

PLOS One
---

## [Editor Report · Acceptance letter]

PONE-D-25-37743R1

PLOS One

Dear Dr. Narisawa,

I'm pleased to inform you that your manuscript has been deemed suitable for publication in PLOS One. Congratulations! Your manuscript is now being handed over to our production team.

Kind regards,

on behalf of

Dr. Pirkko L. Härkönen

Academic Editor

PLOS One